# Quantitative trait loci mapping for salt tolerance-related traits during the germination stage of wheat

**Maoxing Song**[1⊚], **Qing Lu**[1⊚], **Hongliang Ma**[1,2⊚], **Tong Li**[1], **Mengying Yang**[1], **Rongkai Yu**[3], **Huina Huang**[4], **Peng Wu**[5], **Pengjing Liu**[1], **Zhihui Wu**[1]*

**1** Tangshan Academy of Agricultural Sciences, Tangshan, Hebei, China, **2** Chinese Academy of Agricultural Sciences, Beijing, China, **3** Hebei North University, Zhangjiakou, Hebei, China, **4** Guye District Market Supervision Administration of Tangshan, Tangshan, Hebei, China, **5** College of Animal Science and Technology, Hebei Agricultural University, Baoding, Hebei, China

⊚ These authors contributed equally to this work.
* wzh406@sina.cn

## Abstract

Soil salinization is a type of abiotic stress that affects the growth and development of wheat. To explore the QTLs related to salt tolerance during the germination stage of wheat and to reveal the mechanisms of salt tolerance, this study subjected 196 wheat varieties (lines) from North China, East China, and Central China to salt stress during the germination stage. Principal component analysis was employed for a comprehensive evaluation of salt tolerance. Based on the comprehensive evaluation D value, the salt tolerance of the research materials was classified into five levels, from which 64 materials exhibiting salt tolerance or higher were selected. Further, a genome-wide association analysis was conducted on the phenotypic traits and D values of wheat under different treatments during the germination stage using sequencing data from a 16Kb SNP chip. A total of 108 QTLs significantly associated with salt tolerance during the germination stage were identified, distributed across 15 chromosomes, excluding 1A, 1D, 4A, 5B, 6B, and 7B. Individual significant SNPs could explain 8.03% to 22.62% of the phenotypic variation. Additionally, 11 candidate genes potentially related to the salt response in wheat were predicted. This study provides a theoretical basis for the cloning of salt tolerance-related genes and the breeding of salt-tolerant wheat varieties.

## Introduction

Soil salinization is a serious problem faced by modern agriculture, posing a direct threat to food security and sustainable agricultural development [1–3]. According to the Global Map of Salt-affected Soils (GSASmap) provided by Food and Agriculture Organization (FAO), with the current information from 118 countries covering 85% of global land area, it shows that more than 424 million hectares of topsoil (0–30 cm) and 833 million hectares of subsoil (30–100 cm) are salt-affected [4]. In China, saline-alkali land accounts for about 10% of the

**Data availability statement:** All relevant data are within the paper and its Supporting Information files.

**Funding:** We acknowledge the support from various projects, including the Hebei Province Major Science and Technology Support Program Project: Innovation, Integration, and Application of Technology Models for Capacity Enhancement in Coastal Saline-Alkali Areas (242N6401Z) received by Mr. Zhihui Wu; the Hebei Province Modern Agricultural Industry Technology System Drought-Alkali Wheat Innovation Team-Coastal Saline-alkali Crop Comprehensive Experiment and Promotion Station (HBCT2024030404) received by Mr. Zhihui Wu; the Tangshan City Science and Technology Program Projects (23150201A, 23150204A, 24150204C) received by Mr. Maoxing Song; and then Hebei Province Modern Agricultural Industry Technology System Wheat Innovation Team—Tangshan Rice Experimental Station (HBCT2024010408) received by Mr. Hongliang Ma.

**Competing interests:** The authors have declared that no competing interests exist.

total arable land area. The phenomenon of salinization is worsening due to multiple factors, including inappropriate irrigation practices, increasing industrial pollution, and global climate change [5–7]. One effective measure for the comprehensive utilization of saline-alkali soils is the breeding and promotion of salt-tolerant crop varieties. Wheat (*Triticum aestivum* L.) is one of the main food crops in the world [8–10], and cultivating salt-tolerant wheat varieties can improve the utilization of saline-alkali soils and increase overall food production [11,12].

In China, saline-alkali land is extensively distributed in the regions of North China, East China, and Central China. Among these regions, Shandong, Hebei, and Shanxi have the largest areas of saline-alkali land, covering 476,000 hectares, 388,900 hectares, and 300,000 hectares, respectively. The salt-tolerant wheat varieties suitable for cultivation in these regions mainly include Cangmai 6005, Shinong 086, Hengguan 35, Jimai 1, Jinmai 44, Heshangmai, and Jimai 262. However, there are issues such as a limited number of salt-tolerant wheat varieties and slow updates. Therefore, conducting salt tolerance research on wheat varieties originating from North China, East China, and Central China will be beneficial for breeding salt-tolerant wheat varieties suitable for cultivation and promotion in these regions [13].

Identifying salt-tolerant wheat germplasm resources is fundamental for understanding the salt tolerance mechanisms of wheat and breeding salt-tolerant wheat varieties [14–16]. Currently, the evaluation methods for wheat salt tolerance are diverse, with different scholars adopting various evaluation methods. Among these, principal component analysis (PCA) is a widely recognized analytical method that can simultaneously utilize multiple indicators to comprehensively reflect salt tolerance and the interactions between the indicators [17].

At present, the analysis of the molecular mechanisms of wheat salt tolerance is still in its early stages. With the development of sequencing technology, next-generation quantitative trait locus (QTL) mapping techniques represented by genome-wide association studies (GWAS) have become powerful tools for analyzing the genetics of complex traits due to their speed, efficiency, and accuracy [18,19]. These techniques provide precise and efficient pathways for revealing the molecular mechanisms of wheat salt tolerance and accelerate the molecular breeding process for salt-tolerant wheat varieties [20]. Wang et al. [21] conducted principal component analysis on the salt tolerance traits of 259 spring wheat materials during the germination stage and further combined high-density single nucleotide polymorphism (SNP) chips to detect 12 SNP loci associated with wheat salt tolerance distributed on chromosomes 1A, 1B, 2A, 3A, 4A, 4B, 5A, 5B, 6A, and 7A. Kunduzayi et al. [22] used 205 winter wheat varieties to perform genome-wide association analysis, detecting 109 loci associated with wheat salt tolerance on 15 chromosomes, excluding chromosomes 2A, 4A, 4B, 5D, 6D, and 7D. Additionally, Wang et al. [23] located the QTL for wheat salt tolerance on chromosomes 3A, 4A, 7A, and 7B using a recombinant inbred line (RIL) population consisting of 184 lines derived from the cross Tainong 18 × Linmai 6.

The germination stage of wheat marks the beginning of its life cycle [24] and lays the foundation for the entire growth and development process, making it a key stage for in-depth research on crop salt tolerance [25,26]. This study is based on the current situation of wheat production on saline-alkali soils in North China, East China, and Central China. This research subjected 196 wheat varieties from these regions to salt stress treatment during the germination stage and utilized principal component analysis for a comprehensive evaluation of their salt tolerance. Additionally, a 16Kb high-density SNP chip for wheat was combined to conduct genome-wide association analysis on wheat germination stage salt tolerance, with the

aim of identifying QTL loci related to salt tolerance during germination. This study will lay the foundation for cloning salt tolerance-related genes and breeding salt-tolerant varieties of wheat.

## Materials and methods

A total of 196 winter wheat varieties (lines) were sourced from North China, East China, and Central China (S1 Table).

In each replication of the experiment, 30 healthy and uniform wheat seeds were selected, which were disinfected with a 10% NaClO solution for 10 minutes and rinsed twice with distilled water. After disinfection, the seeds were placed in 90 mm Petri dishes lined with sterile filter paper with the embryo side facing down. For the treatment groups, 20 mL of a 1.2% NaCl solution was added to each Petri dish, while the control group received 20 mL of distilled water; each treatment was replicated three times. The seeds were soaked for 4 hours to allow them to imbibe fully, after which 8 mL of solution was removed from each Petri dish. The Petri dishes were then placed in an incubator set at a constant temperature of 20°C, with a light cycle of 16 hours of light and 8 hours of darkness.

According to the Chinese national standard GB/T3543.4-1995 (Seed Testing Procedures for Crops), germination potential and germination rate were assessed at 4 days and 8 days, respectively. The seed germination criteria are as follows: the length of the germ (embryo) should be greater than 50% of the seed length, and the root length should be greater than or equal to the seed length. At 8 days, 10 seedlings were randomly selected for measurement of relevant indicators, including root number (RN), root length (RL), shoot length (SL), shoot fresh weight (SFW), and root fresh weight (RFW).

Based on the statistical results, the following were calculated: germination potential (GP), germination rate (GR), relative germination potential (RGP), relative germination rate (RGR), relative root number (RRN), relative root length (RRL), relative shoot length (RSL), relative root fresh weight (RRFW), relative shoot fresh weight (RSFW), and relative salt injury rate (RSIR). The calculation formulas are as follows:

$$\text{Germination potential GP }(\%) = (\text{Number of germinated seeds at 4 days / Total number of seeds}) \times 100\% \quad (1)$$

$$\text{Germination rate GR }(\%) = (\text{Number of germinated seeds at 8 days / Total number of seeds}) \times 100\% \quad (2)$$

$$\text{Relative values }(\%) = (\text{Value of treated index / Value of control index}) \times 100\% \quad (3)$$

$$\text{Relative salt injury rate }(\%) = [(\text{Control germination rate} - \text{Treatment germination rate}) / \text{Control germination rate}] \times 100\% \quad (4)$$

Data were statistically analyzed using Microsoft Office Excel 2016.

The overall computation process is shown in Fig 1. A total of 22 traits, including those from the control group, salt treatment group, and relative values, were used as original indicators. Based on the analysis method by Peng et al. [27], principal component analysis (PCA) was performed on these original indicators. When the cumulative contribution rate exceeded 90%, the corresponding principal components were used as comprehensive indicators for evaluating salt tolerance. The D-value for salt tolerance was further calculated through a comprehensive evaluation.

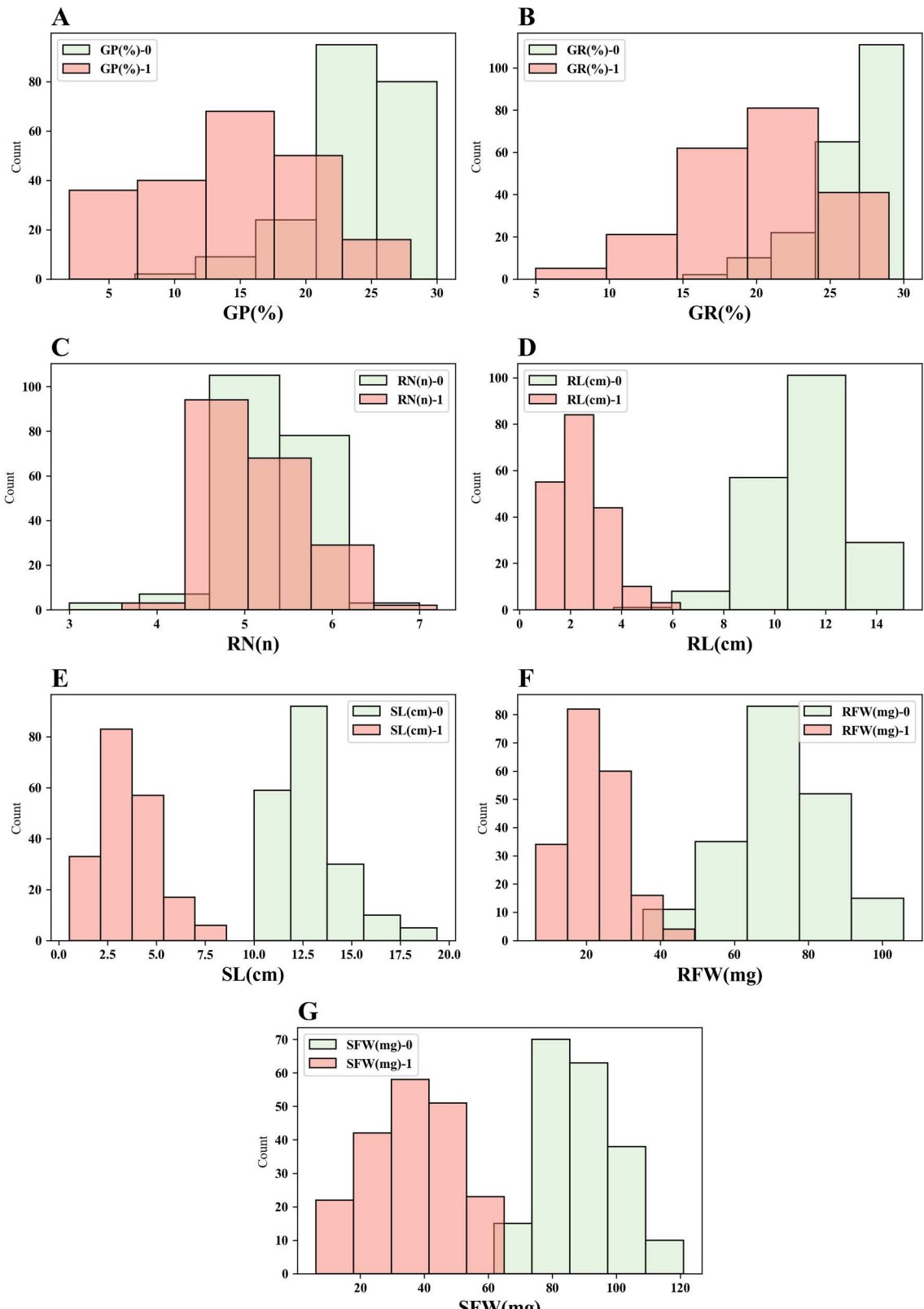

**Fig 1. Overlaid histograms showing frequency distributions of GP (A), GR (B), RN (C), RL (D), SL (E), RFW (F), SFW (G) across control (green) and salt treatment (red).**

K-Means clustering analysis was applied to the comprehensive evaluation of D-values, categorizing them into five classes ranging from high tolerance (HT), tolerant (T), moderately tolerant (MT), sensitive (S), and highly sensitive (HS) in descending order of salt tolerance levels. The above analyses were primarily conducted using the Scikit-learn toolkit in the Python development environment.

Using the 16Kb high-density SNP chip from Shijiazhuang Boruide Biotechnology Co., Ltd., we performed genotyping on a natural population composed of 196 winter wheat varieties (lines). After quality control, we removed samples with a detection rate of loci < 90%, low-frequency genotype frequencies (MAF) < 5%, and SNP markers with over 30% missing data. The final high-quality samples and high-quality SNP markers were utilized for genome-wide association analysis [28].

Population structure analysis was conducted using Structure 2.3.4 software, and the results were analyzed with the online tool Structure Harvester (https://taylor0.biology.ucla.edu/structureHarvester/) to calculate ΔK and determine the number of subpopulations. Population structure plotting was carried out using CLUMPP and GraphPad Prism software.

Linkage disequilibrium (LD) analysis was performed using RTM-GWAS software, which employs a restrictive two-phase multi-locus GWAS method. The results of LD decay distance were visualized using packages such as pandas and seaborn in the Python development environment.

We used the mixed linear model (MLM) of the TASSEL 5.0 analysis software, a genome-wide association analysis was conducted on high-quality samples and SNP marker information after quality control. This analysis incorporated the phenotypic traits, relative values, relative salt injury rates, and comprehensive evaluation D values of salt tolerance for wheat under various treatments during the germination stage, with a threshold set at $p < 0.0001$. Relevant plotting was performed using visualization packages such as geneview, pandas, and seaborn in the Python development environment.

## Results

Under salt stress conditions, various indicators of wheat, including germination potential and germination rate, were reduced compared to the control group (Table 1, Fig 1). During the germination phase under salt stress, root length and shoot length decreased by over 70%, while root fresh weight and shoot fresh weight decreased by more than 50%. Germination

**Table 1. Phenotypic analysis of various indicators during the germination stage under salt stress in wheat** △T-CK represents the difference between the treatment group and the control group.

|  | Index | GP (%) | GR (%) | RN | RL (cm) | SL (cm) | RFW (%) | SFW (%) |
|---|---|---|---|---|---|---|---|---|
| CK | Max | 100 | 100 | 7 | 15.1 | 19.4 | 105.8 | 121.1 |
|  | Min | 23.3 | 50 | 3 | 3.7 | 10 | 35.3 | 61.8 |
|  | Average | 80.1 | 87.8 | 5.2 | 11.1 | 12.9 | 72.3 | 88.9 |
|  | CV (%) | 17 | 11.5 | 9.8 | 15.6 | 13.3 | 18.3 | 13.5 |
| 1.2% NaCl (T) | Max | 93.3 | 96.7 | 7.2 | 6.3 | 8.6 | 49.3 | 65 |
|  | Min | 6.7 | 16.7 | 3.6 | 0.6 | 0.5 | 6.3 | 6.1 |
|  | Average | 47.4 | 67 | 5.2 | 2.5 | 3.6 | 22.5 | 36.2 |
|  | CV (%) | 43.8 | 24 | 8.9 | 41.2 | 44.3 | 33.8 | 37.9 |
| △T-CK | Average | -32.6 | -20.8 | -0.1 | -8.6 | -9.3 | -49.8 | -52.7 |
| △(T-CK)/CK (%) | Relative average | -40.8 | -23.7 | -1.9 | -77.5 | -72.1 | -68.9 | -59.3 |
| Broad-sense heritability (%) |  | 29.31 | 44.73 | 42.84 | 12.26 | 40.88 | 32.81 | 42.94 |

potential decreased by 40.8%, and germination rate decreased by 23.7%, while root number decreased by only 1.9%. The results indicate that salt stress has a significant inhibitory effect on seed germination and the growth of roots and shoots, although it does not noticeably affect the number of roots.

As shown in Table 1, due to the varying sensitivity of different indicators to salt stress, re-lying solely on a single indicator makes it difficult to comprehensively assess the salt tolerance of wheat. Therefore, employing a multi-indicator comprehensive analysis method allows for a more scientific and rational evaluation of wheat's salt tolerance. In this study, principal component analysis (PCA) was conducted on the salt tolerance indicators at the germination stage of 196 wheat materials. Using a cumulative contribution rate threshold of 90%, four independent components (PC1, PC2, PC3, PC4) were selected for comprehensive evaluation of salt tolerance, with contribution rates of 54.383%, 18.491%, 15.171%, and 6.480%, respectively (Table 2). In the principal components analysis, certain indicators with larger absolute values significantly influenced each principal component. Specifically, relative germination potential, relative germination rate, and relative salt injury rate were the main contributors to PC1; relative germination potential and relative shoot fresh weight were the main contributors to PC2; relative germination rate and relative salt injury rate were the primary contributors to PC3; and relative root number was the main contributor to PC4.

The four obtained principal components were used to comprehensively evaluate the salt tolerance of wheat at the germination stage. The membership function values were analyzed to evaluate the comprehensive D-value, and a correlation analysis was conducted on a total of 22 original indicators related to wheat salt tolerance, including the control group, salt treatment group, and relative values (Fig 2). The results indicated a significant correlation between the comprehensive D-value and each original indicator. The strongest correlations were observed with the relative salt injury rate (-0.92), relative germination rate (0.92), and germination rate under salt stress (0.84).

By comparing the correlation coefficients of the comprehensive D-value with various original indicators under different treatments (control group, salt treatment group, relative values) (Fig 3), we observed that the comprehensive D-value generally showed the strongest correlation with the relative values, followed by the salt stress group, while the correlation with the control group was weaker. Among the different traits, the indicators most strongly correlated with the comprehensive D-value were, in order: germination rate, germination potential,

**Table 2. The analysis results of the eigenvalues and contribution rates of the four principal components, as well as the eigenvector analysis of each salt tolerance related indicator.**

| Index | PC1 | PC2 | PC3 | PC4 |
|---|---|---|---|---|
| Eigenvalue | 0.112 | 0.038 | 0.031 | 0.013 |
| Contribution (%) | 54.383 | 18.491 | 15.171 | 6.48 |
| Cumulative contribution (%) | 54.383 | 72.874 | 88.045 | 94.526 |
| RGR | 0.408 | 0.158 | 0.552 | -0.031 |
| RGP | 0.751 | -0.506 | -0.417 | -0.071 |
| RRL | 0.064 | 0.365 | -0.244 | -0.212 |
| RSL | 0.153 | 0.422 | -0.21 | -0.031 |
| RRN | 0.098 | 0.062 | -0.057 | 0.969 |
| RRFW | 0.136 | 0.34 | -0.221 | -0.075 |
| RSFW | 0.216 | 0.513 | -0.249 | 0.058 |
| RSIR | -0.408 | -0.158 | -0.552 | 0.031 |

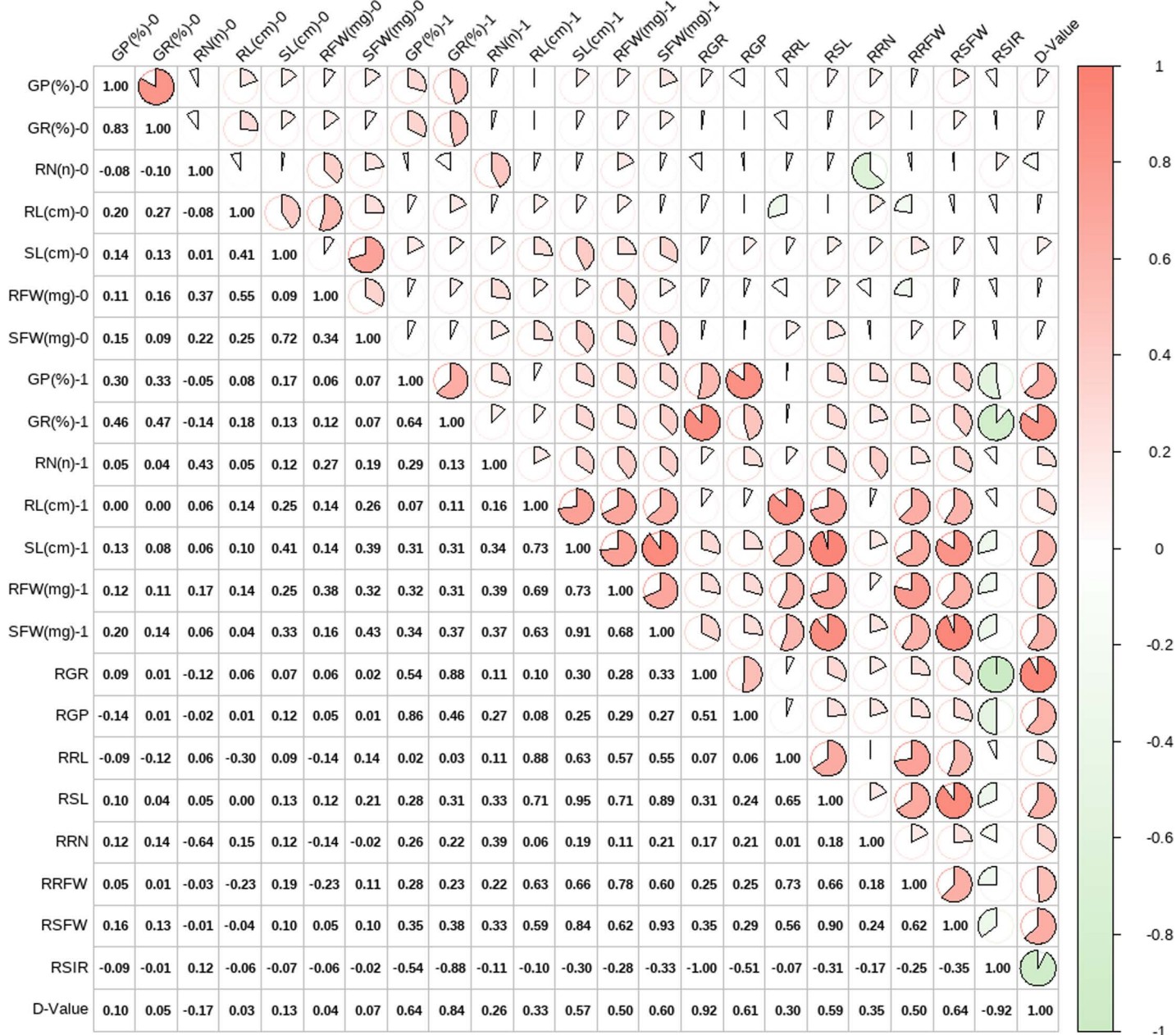

**Fig 2. Correlation analysis of salt tolerance indicators during germination stage in wheat (0 indicates the control group, while 1 indicates the salt treatment group).**

shoot fresh weight, shoot length, root fresh weight, root number, and root length. This indicates that traits related to shoots had a greater correlation than those related to roots.

In summary, the comprehensive evaluation D-value shows a certain correlation with each original indicator and can more comprehensively reflect the salt tolerance of wheat at the germination stage.

Using the comprehensive evaluation D-value, we employed the K-Means clustering method to classify the salt tolerance of the studied materials into five levels (Fig 4): high tolerance (HT), tolerant (T), moderately tolerant (MT), sensitive (S), and highly sensitive (HS).

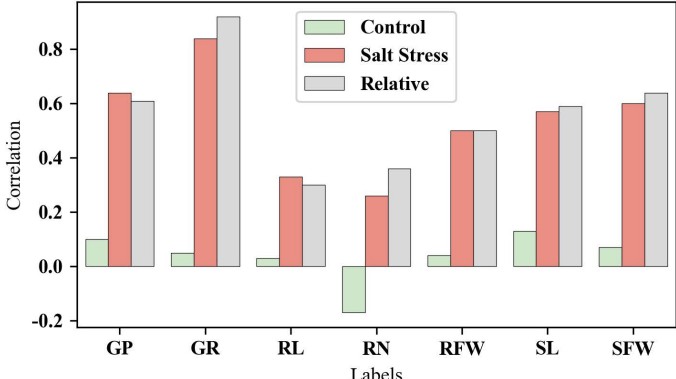

**Fig 3. Correlation coefficients between the comprehensive evaluation D-value and original indicators.**

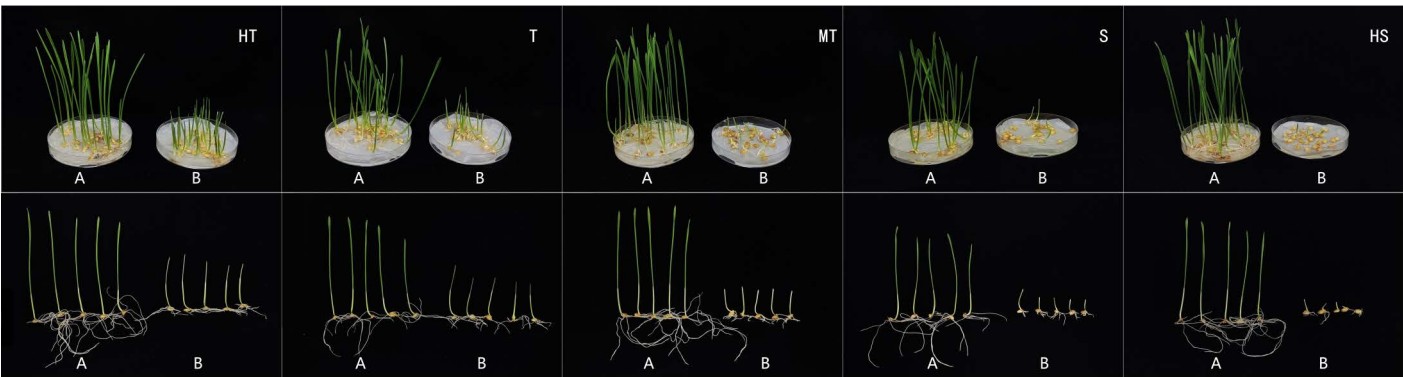

**Fig 4. Salt tolerance grade at germination stage in wheat. HT (high tolerance, Jinmai 70), T (tolerance, Nongda 399), MT (medium tolerance, Liangxing 66), S (sensitivity, Ruika 288) and HS (high sensitivity, Jinghua 11). A and B respectively represent the control and treatment with 1.2% NaCl solution for salt stress.**

The distribution plot on PC1 and PC2 (Fig 5) indicates a good clustering effect among the five categories.

Through the analysis, a total of 17 high-tolerance varieties were screened, including Yanong 0428, Jinmai 70, Shixin 828, Womai U876, Jinmai 59, and Yunhei 161. Additionally, 47 tolerant varieties were identified, including Nongda 399, Heng 05-6607, Cangmai 2016-8, Jinhao 13294, and Demai 1201. Furthermore, 73 moderately tolerant varieties were selected, including Liangxing 66, Ji 5265, Jizi Mai 20, Jinhao 14219, Shimai 16, Zhenghan 36, and Jiemai 20.

We used the wheat 16Kb SNP chip, a total of 14,868 core SNP markers were detected in the population. After phenotypic screening and comprehensive quality control, 196 high-quality samples and 9,501 high-quality core SNP markers were ultimately obtained for the genome-wide association analysis of wheat salt tolerance at the germination stage (Fig 6A).

Population structure analysis was performed using Structure 2.3.4 software, and when K = 3, the ΔK value was maximized, leading to the classification of the population into three subpopulations (Fig 6B and 6C).

We used the quality-controlled high-quality SNP data, LD decay distance was analyzed and plotted (Fig 6D). Following the method for calculating LD decay distance described by Zhao et al. [29–31], the results indicated that when $R^2$ decayed to 50% of its maximum value, the LD decay distances for the A, B, and D subgenomes were in the order A > B > D, with the overall

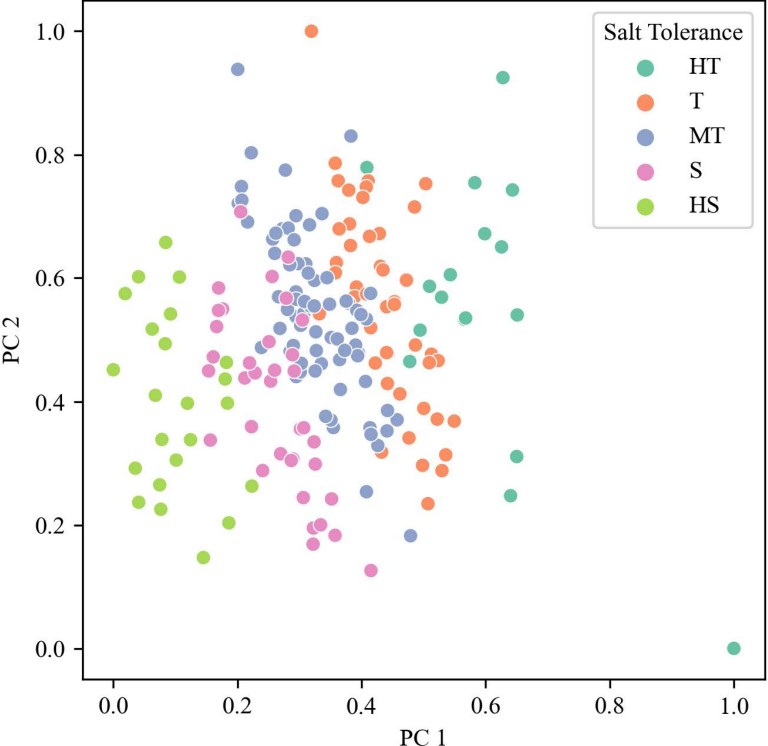

**Fig 5. Cluster analysis of salt tolerance at germination stage in wheat.**

genome's LD decay distance approximately 8 Mb. Thus, significant loci within 8 Mb were considered to be in the same QTL interval.

This study conducted a comprehensive evaluation of salt tolerance in the germination stage of 196 wheat samples and performed GWAS analysis combined with 16K chip data, identifying 199 significant SNP markers. Based on the calculation of LD decay distances, significant markers within 8 Mb were considered to belong to the same QTL interval, resulting in the discovery of 108 QTLs significantly associated with salt tolerance during the germination stage of wheat. These QTLs are distributed across 15 chromosomes, excluding 1A, 1D, 4A, 5B, 6B, and 7B, with individual significant SNPs explaining 8.03% to 22.62% of the phenotypic variation.

The distribution of these QTLs and SNP markers in the wheat genome is significantly different, with the highest number of QTLs located on chromosome 4B, totaling 43, which includes 72 SNP markers. Following that, chromosome 3B contains a total of 13 QTLs with 31 SNP markers. Additionally, chromosome 3D also has a relatively high number of QTLs, totaling 12, which includes 18 SNP markers. Chromosome 1B contains 10 QTLs, totaling 43, SNP markers. In contrast, chromosome 2A has slightly fewer QTLs, with a total of 8, including 11 SNP markers. The numbers of QTLs on chromosomes 2B and 6A are the same, with each having 4 QTLs, corresponding to 4 and 6 SNP markers, respectively. Chromosome 6D has the fewest QTLs, with only 3, which include 3 SNP markers. Furthermore, chromosomes 2D, 3A, 4D, and 7D each have 2 QTLs, corresponding to 2 SNP markers. Finally, chromosomes 5A, 5D, and 7A have the least number of QTLs, with only 1 QTL per chromosome, each containing 1 SNP marker (Figs 7 and 8, S2 Table).

In the association analysis between traits and significant QTL loci, we found that all traits except for GR and RSIR successfully associated with their corresponding QTLs. Specifically,

**A** Illumilla 16K

**C**

$$DeltaK = mean(|L'''(K)| / sd(L(K))$$

Fig 6. SNP site distribution (A), Population structure (B), K-value (C), and LD decay distance (D).

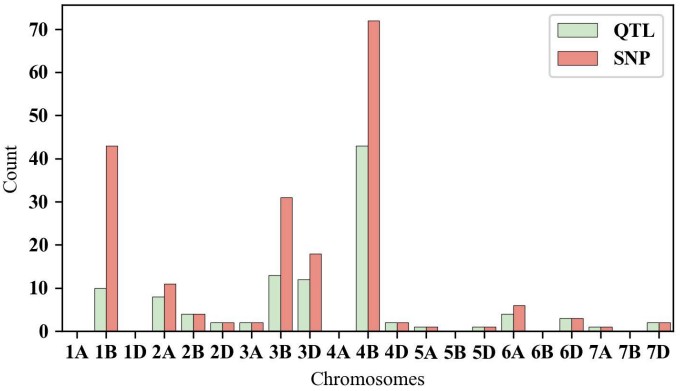

Fig 7. Chromosome-wide distribution of QTL (green) and SNP (red).

the GP trait was particularly prominent, identifying the highest number of QTLs, totaling 36, among which GP_0 and GP_1 were associated with 2 QTLs each, while RGP had the most, with 32 QTLs. Next, D_Value was associated with 32 QTLs. The RN trait was associated with 28 QTLs, with further subdivision revealing that RN_0, RN_1, and RRN were associated with 1, 20, and 7 QTLs, respectively. In contrast, the number of QTLs associated with the RFW trait was relatively lower, with only RFW_1 linking to 4 QTLs. As for the RL trait, there were a total of 3 QTLs, with RL_1 containing 1 and RRL containing 2. Additionally, the SFW and SL traits were also associated with a certain number of QTLs under control treatment conditions, with SFW_0 associated with 3 QTL loci and SL_0 associated with 2 QTLs (Fig 8, S2 Table).

Using the Ensemble Plants tool (https://plants.ensembl.org/Triticum_aestivum/Info/Index) to search for candidate genes within the QTL intervals with high phenotypic variance, a total of 175 predicted genes were identified. Further functional comparison of these 175 predicted genes was conducted using the NCBI search tool (https://www.ncbi.nlm.nih.gov/), resulting in the selection of 11 potential candidate genes that may be associated with plant salt tolerance responses (Table 3). These genes could be involved in the salt tolerance response process of wheat. Among them, *TraesCS2A02G093700* encodes a type 2C protein phosphatase; *TraesCS2A02G095300* encodes a serine/threonine protein kinase; *TraesCS3B02G493700* encodes a protein containing a psbP domain, found in chloroplasts; *TraesCS3B02G497200* encodes a zinc finger protein; *TraesCS4B02G202200* encodes a membrane-associated kinase regulator; *TraesCS4B02G207400* encodes an E3 ubiquitin-protein ligase; *TraesCS4B02G215800* encodes an ethylene response transcription factor; *TraesCS4B02G216700* encodes a mitogen-activated protein kinase; *TraesCS7D02G015100* encodes an F-box protein; *TraesCS7D02G016800* encodes a ferredoxin-NADP + reductase, a leaf isoenzyme located in plastids; and *TraesCS7D02G018900* encodes a disease resistance protein.

## Discussion

The entire growth cycle of wheat includes several developmental stages such as germination, seedling, and jointing stages, and the salt tolerance at any given stage cannot represent its salt tolerance throughout the entire growth period. Therefore, combining indoor and field testing methods for evaluation ensures a more comprehensive assessment of wheat's salt tolerance. The germination stage is a critical phase for assessing salt tolerance in a controlled environment, and the performance in this stage is essential for evaluating the overall true salt tolerance.

In this study, analysis of various indicators of wheat at the germination stage under salt stress conditions revealed that salt stress has the most significant impact on root length and shoot length, with root fresh weight and shoot fresh weight being affected to a lesser extent. Among these, root length and root fresh weight were slightly more affected than shoot length and shoot fresh weight. These findings are consistent with those of Wang et al. [21] and Peng et al. [27]. However, a key difference is that in this study, the number of roots was the least affected by salt stress, while in the studies of Wang and Peng, the variation in root number reached significant levels. The differences in results may be due to variations in experimental materials and conditions.

Given that different indicators exhibit substantial differences in sensitivity to salt stress conditions, a single indicator cannot comprehensively evaluate wheat's salt tolerance. Therefore, employing multiple indicators for a comprehensive analysis using Principal Component Analysis (PCA) can provide a more scientific and rational evaluation of wheat's

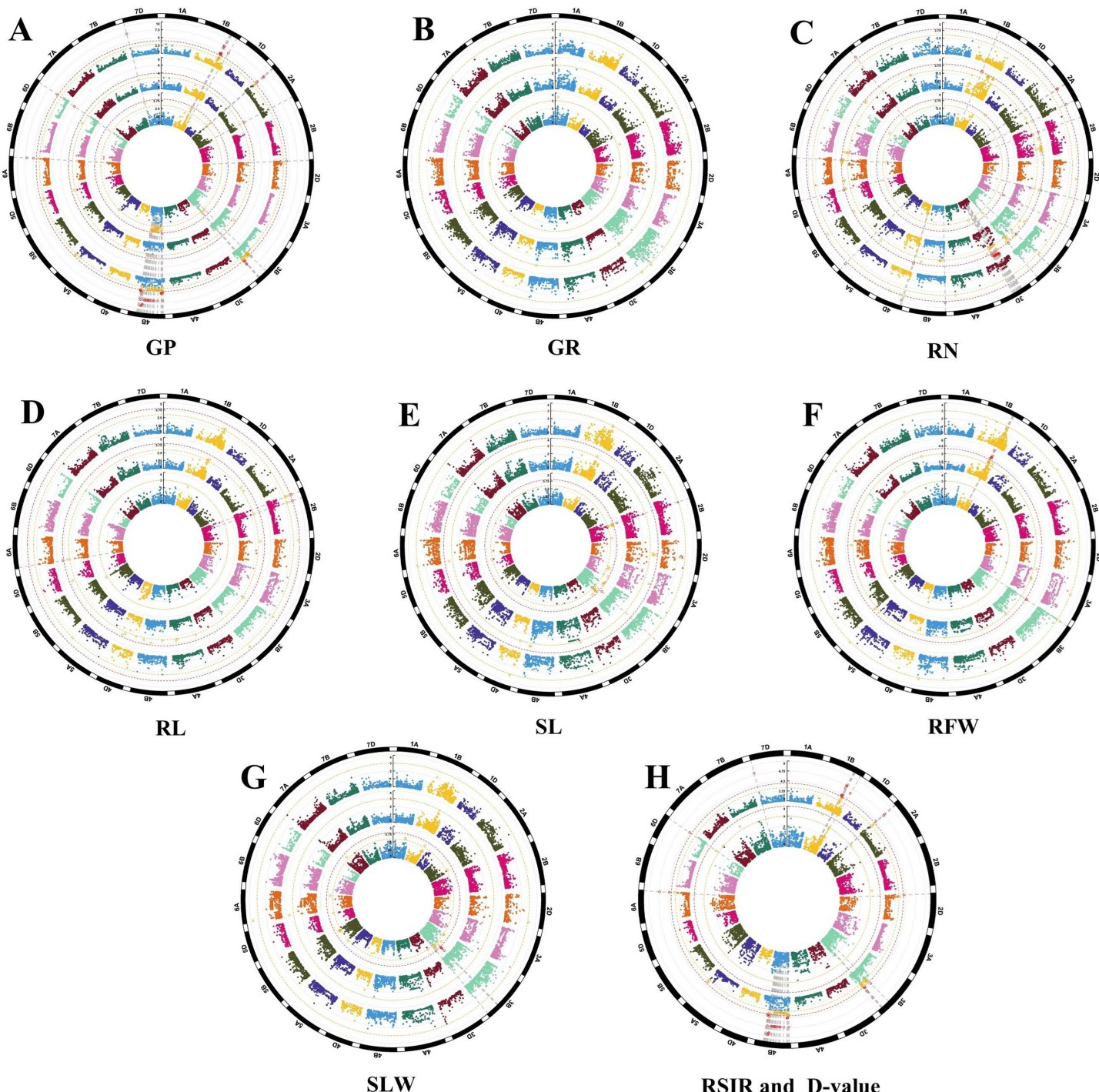

**Fig 8. Manhattan plot of GWAS of GP (A), GR (B), RN (C), RL (D), SL (E), RFW (F), SFW (G) RSIR (H Inner circle) and D-value(H outer circle).** A–G represent the control group, salt treatment group, and relative values from innermost circle to outermost circle.

salt tolerance, which has now become one of the mainstream methods for comprehensive evaluation. In this study, PCA was applied to identify four principal components with a cumulative contribution rate of 94.526%. These components represent the majority of the indicator information and comprehensively reflect the salt tolerance characteristics of wheat

**Table 3. Candidate gene information within the major QTL.**

| QTL | Chromosome | Gene | Gene interval | Gene annotation or coding protein |
|---|---|---|---|---|
| *qSalT-2A45* | 2A | *TraesCS2A02G093700* | 47830215 –47833893 | Protein phosphatase 2C |
| | 2A | *TraesCS2A02G095300* | 48855363 – 48865404 | Serine/threonine-protein kinase |
| *qSalT-3B732* | 3B | *TraesCS3B02G493700* | 739024450 – 739027263 | psbP domain-containing protein, chloroplastic |
| | 3B | *TraesCS3B02G497200* | 740543667 – 740546876 | Zinc finger protein GAI-ASSOCIATED FACTOR 1-like |
| *qSalT-4B433* | 4B | *TraesCS4B02G202200* | 432823542 – 432824381 | Probable membrane-associated kinase regulator |
| *qSalT-4B443* | 4B | *TraesCS4B02G207400* | 442586044 – 442589052 | E3 ubiquitin-protein ligase |
| *qSalT-4B453* | 4B | *TraesCS4B02G215800* | 455615441 – 455616792 | Ethylene-responsive transcription factor |
| | 4B | *TraesCS4B02G216700* | 456594135 – 456598490 | Mitogen-activated protein kinase |
| *qSalT-7D7* | 7D | *TraesCS7D02G015100* | 6490876– 6492702 | F-box protein |
| | 7D | *TraesCS7D02G016800* | 7446366 – 7449642 | Ferredoxin-NADP reductase, leaf isozyme, chloroplastic-like |
| | 7D | *TraesCS7D02G018900* | 8343019 – 8347549 | Disease resistance protein |

at the germination stage. By utilizing these four principal components and further analyzing them with membership function values, a comprehensive evaluation of wheat's salt tolerance at the germination stage was conducted. The resulting comprehensive evaluation D-value can be used as a phenotypic value for discovering QTLs related to wheat germination stage salt tolerance, potentially providing more precise results than those obtained using single indicators.

Additionally, some researchers have calculated indicator values (such as germination rate) under different salt concentrations and applied integrated methods to derive corresponding salt tolerance indices. These indices can eliminate the influence of salt concentration factors on the assessment results of wheat's salt tolerance at the germination stage, providing a more accurate reflection of wheat's true tolerance under varying salt stress environments [32]. In future assessments of wheat salt tolerance, combining PCA with an integrated method-based salt tolerance index could enhance the accuracy and comprehensiveness of evaluations. This approach would help mitigate the influence of individual indicators or salt concentrations on experimental results, particularly during the germination stage. Ultimately, it would lead to more scientific, thorough, and precise analysis of wheat salt tolerance.

In this study, we conducted an association analysis between traits and significant QTL loci and found that all traits, except for germination rate (GR) and relative salt injury rate (RSIR), were successfully associated with their corresponding QTLs. The roots play an important role in perceiving salt stress, and the research by Akram et al. [33] revealed the presence of loci associated with root number (RN) on 17 chromosomes, excluding chromosomes 1A, 1B, 1D, and 7B, under different salt concentration stress conditions. Our study also identified key loci controlling RN on chromosomes 2A, 2B, 3A, 3B, 3D, 4B, 4D, 5A, and 5D, while additionally identifying potentially new QTL loci on chromosomes 1B, 5B, 6B, 6D, and 7D. Furthermore, by comparison, we found that the *qSalT_5A702* locus located on chromosome 5A (702 Mb) overlaps with loci related to growth under salt stress discovered by Asif et al. [34]. On the other hand, Guo et al. [35] located important loci controlling root length (RL) on chromosomes 2A, 2B, 3B, and 4A, while our study also detected corresponding QTLs on chromosome 2B. These loci differ from the results of Akram et al. [36], Javid et al. [37] and Khan et al. [38]. In summary, chromosome 2B likely contains key loci regulating various root traits (such as root length and number), which provides important insights for further exploring the salt tolerance mechanisms in wheat roots.

Additionally, previous studies (including Javid et al. [37], Khan et al. [39], and Oyiga et al. [40]) have localized loci controlling root fresh weight (RFW) on chromosomes 2A, 2B, 7A, and 7B. However, our study localized the regulatory loci for RFW on chromosomes 1B and 3B, finding that the *qSalT_3B14* locus located on chromosome 3B (14 Mb) overlaps with loci identified by Pasam et al. [41] that are related to Na + content in the second leaf under salt stress.

In the exploration of the role of the coleoptile in the seed germination process, our study located QTLs controlling shoot length (SL) on chromosomes 2B and 3B. This finding is similar to the reports by Akram et al. and Javid et al., but does not overlap with the results of Guo et al. (2022). Furthermore, we also identified a QTL affecting shoot fresh weight (SFW) located on chromosome 3B, and this conclusion aligns with the findings of Guo et al. [35], Amin et al. [42], and Masoudi et al. [43], but differs from the results of Javid et al. [37] and Oyiga et al. [40], who found SFW-related loci on chromosomes 6A, 6D, and 2D, 3A, and 7B, respectively. In summary, the 3B region may harbor key genes regulating shoot growth-related traits.

Through a comprehensive assessment of seed vigor during germination under different salt concentration stresses, including germination rate (GR), relative salt injury rate (RSIR), germination potential (GP), and D-Value, we can more comprehensively measure the salt tolerance of wheat. However, our study did not identify quantitative trait loci (QTL) associated with GR and RSIR. Guo et al. localized loci associated with GP on chromosomes 2B, 3B, and 6A, while we also found related QTLs on chromosomes 3B and 6A. In the analysis of D-Value, we employed an innovative approach by combining principal component analysis and membership function values to comprehensively evaluate the salt tolerance of wheat in its germination stage, which has been reported less frequently. We located QTLs related to D-Value on chromosomes 1B, 2A, 2D, 3B, 4B, 6A, 6D, and 7D. Notably, the *qSalT_4B389* locus located on chromosome 4B (389 Mb) overlaps with loci related to SPAD values found by Javid et al. at the same position (4B, 389.1 Mb). Additionally, the *qSalT_1B492* locus located on chromosome 1B (492 Mb - 516 Mb) and the *qSalT_6A601* locus located on chromosome 6A (601 Mb) overlap with loci identified by Pasam et al. [41] on chromosome 1B (494 Mb - 497 Mb) and chromosome 6A (600 Mb - 601 Mb) related to the control of K + /Na + distribution in the fourth leaf under salt stress. It is important to note that the *qSalT_1B526* locus located on chromosome 1B (526 Mb - 530 Mb) also overlaps with loci identified by Javid et al. [37], Quamruzzaman et al. [44], and Pasam et al. (2023) on chromosome 1B, which are associated with shoot length (SL), salinity damage score, and K + /Na + distribution in the fourth leaf under salt stress, respectively. This suggests that through an in-depth analysis of D-Value, we may be able to discover some genetic loci controlling traits related to salt stress during the seedling stage of wheat, beyond traditional measurements.

By further comparing the mapping positions of the phenotypic trait-related QTL identified in this study, we found that some QTLs controlling different traits are located close to each other. For instance, the *qSalT_2B172* locus located on chromosome 2B (172 Mb) is associated with relative root length (RRL), while the *qSalT_2B170* locus located on chromosome 2B (170 Mb) is associated with shoot length (SL). This proximity suggests that they may represent the same locus and exhibit pleiotropy. Additionally, the *qSalT_1B489* locus at the position of 489 Mb on chromosome 1B is closely adjacent to the *qSalT_1B492* locus at position 492 Mb on the same chromosome; the former is associated with root fresh weight (RFW), while the latter is an important locus controlling relative germination potential (RGP) and D-Value. This indicates that they may be part of the same genetic locus.

In summary, this study provides valuable insights into the genetic basis of salt tolerance traits in wheat by identifying significant QTLs that could be targeted for breeding programs aimed at enhancing salt resistance. The overlap of identified loci with previously reported loci further strengthens the confidence in these findings and highlights the importance of specific chromosomes, particularly 1B, 2B, and 3B, in regulating various root and shoot traits under salt stress conditions. Further investigation into the functional roles of these QTLs and their associated genes will be essential to develop effective strategies for improving salt tolerance in wheat.

Within plant organisms, salt tolerance genes are generally classified into two categories: regulatory genes and functional genes. Regulatory genes participate in signal transduction and gene expression, such as transcription factors and protein kinases, while functional genes are directly involved in salt tolerance, coding for products that maintain ion homeostasis, detoxifying enzymes, and other functional proteins [45–47]. In this study, we screened 11 potential candidate genes, which belong to the category of regulatory genes upon functional comparison, within the intervals of the 6 relevant QTLs for wheat salt tolerance at the germination stage, all of which belong to the category of regulatory genes upon functional comparison. The gene *TraesCS2A02G093700*, located on chromosome 2A, encodes a type 2C protein phosphatase. Studies have confirmed that type 2C protein phosphatases participate in signaling pathways for abiotic stress/ABA, and regulating the expression of type 2C protein phosphatase genes can significantly enhance wheat salt tolerance [48,49]. The gene *TraesCS2A02G095300* on chromosome 2A encodes a serine/threonine protein kinase, which can improve salt stress tolerance through the regulation of stomatal closure and osmotic adjustments [50]. The gene *TraesCS3B02G497200* on chromosome 3B encodes a zinc finger protein, which plays a role in regulating plant growth and development, responding to stress, and enhancing wheat salt tolerance by mediating responses to salt stress. The gene *TraesCS4B02G207400* on chromosome 4B encodes an E3 ubiquitin-protein ligase, which can precisely identify and tag specific target proteins. Through the ubiquitination pathway, these proteins can be controlled, and when this gene is overexpressed, the salt tolerance of wheat seedlings is improved [51]. Located on chromosome 4B, *TraesCS4B02G216700* encodes a mitogen-activated protein kinase (MAPK), which plays a role in signal transduction between the cell membrane and the nucleus, participating in the transmission of stress signals such as salt stress [52]. The gene *TraesCS7D02G015100*, located on chromosome 7D, encodes an F-box protein, which has been shown to play an important role in the salt stress response in various plants. In Arabidopsis, the F-box protein regulates salt tolerance by modulating the activity of plasma membrane Na + /H + antiporters [53,54]. Additionally, although the remaining 5 candidate genes have been reported preliminarily [55,56], research on them is still insufficient compared to the six candidate genes mentioned above. In order to fully elucidate the specific roles and contributions of these genes in the salt tolerance mechanism of wheat, more systematic and detailed studies are urgently needed in the future.

## Conclusion

In this study, we conducted a comprehensive evaluation of the salt tolerance of 196 wheat varieties (lines) in the germination stage using principal component analysis. We identified 64 materials that exhibited above-average salt tolerance. Further, we performed a genome-wide association analysis on wheat germination stage salt tolerance and discovered 108 QTLs significantly associated with this trait, distributed across 15 chromosomes, excluding chromosomes 1A, 1D, 4A, 5B, 6B, and 7B. Individual significant SNPs explained 8.03% to 22.62% of the phenotypic variation. Additionally, we predicted 11 potential genes that may be related to the plant's salt tolerance response. This research provides a theoretical basis for the cloning of salt tolerance-related genes in wheat and the breeding of salt-tolerant varieties.

## Supporting information

**S1 Table. 196 varieties of wheat information.**
(DOCX)

**S2 Table. List of QTL associated with salinity tolerance component traits in wheat during germination.**
(XLSX)

## Author contributions

**Conceptualization:** Maoxing Song.

**Data curation:** Hongliang Ma.

**Formal analysis:** Tong Li.

**Funding acquisition:** Mengying Yang.

**Investigation:** Rongkai Yu.

**Methodology:** Huina Huang.

**Project administration:** Peng Wu.

**Resources:** Pengjing Liu.

**Software:** Zhihui Wu.

**Supervision:** Zhihui Wu.

**Validation:** Qing Lu.

**Visualization:** Qing Lu.

**Writing – original draft:** Maoxing Song.

**Writing – review & editing:** Maoxing Song.

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
