## [Decision Letter · Decision Letter 0]

29 Dec 2024

PONE-D-24-57089Quantitative Trait Loci Mapping for Salt Tolerance-Related Traits during the Germination Stage of WheatPLOS ONE

Dear Dr. Wu,

Thank you for submitting your manuscript to PLOS ONE. After careful consideration, we feel that it has merit but does not fully meet PLOS ONE’s publication criteria as it currently stands. Therefore, we invite you to submit a revised version of the manuscript that addresses the points raised during the review process.

We look forward to receiving your revised manuscript.

Kind regards,

Aimin Zhang, Ph.D.

Academic Editor

PLOS ONE

Journal Requirements:

The work was funded by a number of projects, including the Hebei Province Major Science and Technology Support Program Project: Innovation, Integration, and Application of Technology Models for Capacity Enhancement in Coastal Saline-Alkali Areas (242N6401Z);Hebei Province Modern Agricultural Industry Technology System Drought-Alkali Wheat Innovation Team-Coastal Saline-alkali Crop Comprehensive Ex-periment and Promotion Station (HBCT2024030404); Tangshan City Science and Tech-nology Program Projects (23150201A, 23150204A, 24150204C); Hebei Province Modern Agricultural Industry Technology System Wheat Innovation Team — Tangshan Rice Ex-perimental Station (HBCT2024010408). 

5. Please include a copy of Table 4 which you refer to in your text on page 38.

Reviewers' comments:

Reviewer's Responses to Questions

**Comments to the Author**

1. Is the manuscript technically sound, and do the data support the conclusions?

Reviewer #1: Yes

Reviewer #2: Yes

2. Has the statistical analysis been performed appropriately and rigorously? 

Reviewer #1: Yes

Reviewer #2: Yes

3. Have the authors made all data underlying the findings in their manuscript fully available?

Reviewer #1: Yes

Reviewer #2: Yes

4. Is the manuscript presented in an intelligible fashion and written in standard English?

Reviewer #1: Yes

Reviewer #2: Yes

5. Review Comments to the Author

Reviewer #1: Soil salinization can affect the growth and development of wheat, and the QTL for salt stress tolerance is of great value in the genetic improvement of wheat stress tolerance in molecular breeding programs. This study reported 22 QTLs significantly associated with salt tolerance in wheat germination stage, and 175 annotated genes were discovered in the six major QTLs. This is a useful study to some extent. My major concerns are as follows:

1. Figure 1 makes no sense and could be removed from the text.

2. The broad-sense heritability of the corresponding phenotypic trait should be supplemented, if possible.

3. Figure 3, the title above the Figure should be removed, as the corresponding information can be seen in the legend.

4. Figure 4, the cultivar names of the corresponding representative samples should be supplemented, if possible.

5. Only D-value was used for GWAS. I suppose to perform GWAS for both D-values and the phenotypic traits under different treatments. Please also compare their mapping position and discover more useful information.

6. Please compare the QTL with that of previous studies based on their physical position or common markers, and determine the novel QTL that were firstly reported in this study. In addition, are there common QTL for yielding potential that co-located with the salt stress tolerance?

7. Overall, the English writing, the whole chart/table layout should be improved.

Reviewer #2: This manuscript “Quantitative Trait Loci Mapping for Salt Tolerance-Related Traits during the Germination Stage of Wheat” by Song et al. This study investigates the genetic basis of salt tolerance in wheat during the germination stage by analyzing 196 wheat varieties from China. The study evaluated salt tolerance traits using principal component analysis (PCA) and categorized the varieties into five tolerance levels. A genome-wide association study (GWAS) with a 16Kb SNP chip identified 22 significant QTLs across six chromosomes (1B, 2A, 3B, 4B, 6A, and 7D), and gene annotation revealed 11 potential candidate genes involved in stress response mechanisms. The findings provide a theoretical foundation for cloning salt-tolerance genes and breeding salt-tolerant wheat varieties to address agricultural challenges in saline-alkaline soils. However, certain aspects require refinement before this can be considered ready for publication:

Comment 1. In the introduction, Please simplify repetitive phrasing by removing “Province” after each name, and replacing “measuring” with “covering” for better word choice.

Comment 2. In the discussion, improving influence by rephrasing “cloning genes related to salt tolerance” as “cloning salt tolerance-related genes”.

Comment 3. Add units to the header of table 1 (For example: "root length (cm)").

Comment 4. Add references to paragraph 3 on page 25.

Comment 5. It is recommended that a figure legend and table legend be added.

Comment 6. Note the format of the figures and tables.

Comment 7. Please make additional careful revisions to the formatting of the manuscript.

6. PLOS authors have the option to publish the peer review history of their article (what does this mean? ). If published, this will include your full peer review and any attached files.

**Do you want your identity to be public for this peer review?** For information about this choice, including consent withdrawal, please see our Privacy Policy .

Reviewer #1: No

Reviewer #2: No

---

## [Author Response · Author response to Decision Letter 1]

14 Jan 2025

Part 2 Revision comments from Reviewer 1.

Soil salinization can affect the growth and development of wheat, and the QTL for salt stress tolerance is of great value in the genetic improvement of wheat stress tolerance in molecular breeding programs. This study reported 22 QTLs significantly associated with salt tolerance in wheat germination stage, and 175 annotated genes were discovered in the six major QTLs. This is a useful study to some extent. My major concerns are as follows:

1. Figure 1 makes no sense and could be removed from the text.

Response: Thank you very much. The title above the Figure has been removed.

2. The broad-sense heritability of the corresponding phenotypic trait should be supplemented, if possible.

Response: I have added broad-sense heritability in the last row at the bottom of Table 1.

3. Figure 3, the title above the Figure should be removed, as the corresponding information can be seen in the legend.

Response: The title above the figure has been removed.

4. Figure 4, the cultivar names of the corresponding representative samples should be supplemented, if possible.

Response: I have indicated the names of the respective varieties in the legend section of Figure 4. T(tolerance, Nongda 399), MT(medium tolerance, Liangxing 66), S(sensitivity, Ruika 288) and HS(high sensitivity, Jinghua 11).

5. Only D-value was used for GWAS. I suppose to perform GWAS for both D-values and the phenotypic traits under different treatments. Please also compare their mapping position and discover more useful information.

Response: Thank you. I have added the relevant experimental results, figures, and discussion. In the association analysis between traits and significant QTL loci, we found that all traits except for GR and RSIR successfully associated with their corresponding QTLs. Specifically, the GP trait was particularly prominent, identifying the highest number of QTLs, totaling 36, among which GP_0 and GP_1 were associated with 2 QTLs each, while RGP had the most, with 32 QTLs. Next, D_Value was associated with 32 QTLs. The RN trait was associated with 28 QTLs, with further subdivision revealing that RN_0, RN_1, and RRN were associated with 1, 20, and 7 QTLs, respectively. In contrast, the number of QTLs associated with the RFW trait was relatively lower, with only RFW_1 linking to 4 QTLs. As for the RL trait, there were a total of 3 QTLs, with RL_1 containing 1 and RRL containing 2. Additionally, the SFW and SL traits were also associated with a certain number of QTLs under control treatment conditions, with SFW_0 associated with 3 QTL loci and SL_0 associated with 2 QTLs. Please refer to the results and discussion in the revised manuscript for more supplementary information.

6. Please compare the QTL with that of previous studies based on their physical position or common markers, and determine the novel QTL that were firstly reported in this study. In addition, are there common QTL for yielding potential that co-located with the salt stress tolerance?

Response: I have added references to the research results of Akram, Asif, Guo, Javid, Khan, Oyiga, Pasam, Amin, and Masoudi. Please refer to the discussion section of the article for details.

7. Overall, the English writing, the whole chart/table layout should be improved.

Response: Thank you. The English writing, the whole chart/table layout has been modified and improved.

Part 3 Revision comments from Reviewer 2.

This manuscript “Quantitative Trait Loci Mapping for Salt Tolerance-Related Traits during the Germination Stage of Wheat” by Song et al. This study investigates the genetic basis of salt tolerance in wheat during the germination stage by analyzing 196 wheat varieties from China. The study evaluated salt tolerance traits using principal component analysis (PCA) and categorized the varieties into five tolerance levels. A genome-wide association study (GWAS) with a 16Kb SNP chip identified 22 significant QTLs across six chromosomes (1B, 2A, 3B, 4B, 6A, and 7D), and gene annotation revealed 11 potential candidate genes involved in stress response mechanisms. The findings provide a theoretical foundation for cloning salt-tolerance genes and breeding salt-tolerant wheat varieties to address agricultural challenges in saline-alkaline soils. However, certain aspects require refinement before this can be considered ready for publication:

Comment 1. In the introduction, Please simplify repetitive phrasing by removing “Province” after each name, and replacing “measuring” with “covering” for better word choice.

Response: Thank you, I have removed ‘Province’ and replaced ‘measuring’ with ‘covering’.

Comment 2. In the discussion, improving influence by rephrasing “cloning genes related to salt tolerance” as “cloning salt tolerance-related genes”.

Response: I have made the modification to the discussion by rephrasing ‘cloning genes related to salt tolerance’ as ‘cloning salt tolerance-related genes’

Comment 3. Add units to the header of table 1 (For example: "root length (cm)").

Response: I have added units to the header of Table 1

Comment 4. Add references to paragraph 3 on page 25.

Response: I have added references to paragraph 3 on page 25

Comment 5. It is recommended that a figure legend and table legend be added.

Response: I have added a figure legend and a table legend as recommended.

Comment 6. Note the format of the figures and tables.

Response: Thank you. I have noted the format of the figures and tables.

Comment 7. Please make additional careful revisions to the formatting of the manuscript.

Response: Thank you. I have made additional careful revisions to the formatting of the manuscript.

---

## [Decision Letter · Decision Letter 1]

3 Feb 2025

Quantitative Trait Loci Mapping for Salt Tolerance-Related Traits during the Germination Stage of Wheat

PONE-D-24-57089R1

Dear Dr. Wu,

We’re pleased to inform you that your manuscript has been judged scientifically suitable for publication and will be formally accepted for publication once it meets all outstanding technical requirements.

Kind regards,

Aimin Zhang, Ph.D.

Academic Editor

PLOS ONE

Additional Editor Comments (optional):

Reviewers' comments:

Reviewer's Responses to Questions

**Comments to the Author**

1. If the authors have adequately addressed your comments raised in a previous round of review and you feel that this manuscript is now acceptable for publication, you may indicate that here to bypass the “Comments to the Author” section, enter your conflict of interest statement in the “Confidential to Editor” section, and submit your "Accept" recommendation.

Reviewer #1: All comments have been addressed

Reviewer #2: All comments have been addressed

2. Is the manuscript technically sound, and do the data support the conclusions?

Reviewer #1: Yes

Reviewer #2: Yes

3. Has the statistical analysis been performed appropriately and rigorously? 

Reviewer #1: Yes

Reviewer #2: Yes

4. Have the authors made all data underlying the findings in their manuscript fully available?

Reviewer #1: Yes

Reviewer #2: Yes

5. Is the manuscript presented in an intelligible fashion and written in standard English?

Reviewer #1: Yes

Reviewer #2: Yes

6. Review Comments to the Author

Reviewer #1: Through modifications, the quality of this MS has greatly improved, and all the questions I raised have been fully addressed. Therefore, I recommend accepting this MS for publication. However, there is one point that the authors need to make further revisions. There are many formatting issues with the references cited, such as italicizing the Latin names of species, italicizing the names of genes, capitalization or not of the first letters in the titles of cited references, and abbreviations of the authors' names in cited references, all of which require further revisions.

Reviewer #2: (No Response)

7. PLOS authors have the option to publish the peer review history of their article (what does this mean? ). If published, this will include your full peer review and any attached files.

**Do you want your identity to be public for this peer review?** For information about this choice, including consent withdrawal, please see our Privacy Policy .

Reviewer #1: No

Reviewer #2: No

---

## [Editor Report · Acceptance letter]

PONE-D-24-57089R1

PLOS ONE

Dear Dr. Wu,

I'm pleased to inform you that your manuscript has been deemed suitable for publication in PLOS ONE. Congratulations! Your manuscript is now being handed over to our production team.

Kind regards,

on behalf of

Prof. Aimin Zhang

Academic Editor

PLOS ONE